# Complement in Human Brain Health: Potential of Dietary Food in Relation to Neurodegenerative Diseases

**DOI:** 10.3390/foods12193580

**Published:** 2023-09-26

**Authors:** Yihang Xing, Dingwen Zhang, Li Fang, Ji Wang, Chunlei Liu, Dan Wu, Xiaoting Liu, Xiyan Wang, Weihong Min

**Affiliations:** 1College of Food Science and Engineering, Jilin Agricultural University, Changchun 130118, China; xyh15943188342@163.com (Y.X.); 15104353149@163.com (D.Z.); fangli1014@126.com (L.F.); wangji198644@163.com (J.W.); liuchunlei0709@jlau.edu.cn (C.L.); wudan@jlau.edu.cn (D.W.); liuxiaoting@jlau.edu.cn (X.L.); 2College of Food and Health, Zhejiang A&F University, Hangzhou 311300, China

**Keywords:** C1q, complement pathway, synaptic pruning, microglia, neurodegenerative diseases, dietary food

## Abstract

The complement pathway is a major component of the innate immune system, which is critical for recognizing and clearing pathogens that rapidly react to defend the body against external pathogens. Many components of this pathway are expressed throughout the brain and play a beneficial role in synaptic pruning in the developing central nervous system (CNS). However, excessive complement-mediated synaptic pruning in the aging or injured brain may play a contributing role in a wide range of neurodegenerative diseases. Complement Component 1q (C1q), an initiating recognition molecule of the classical complement pathway, can interact with a variety of ligands and perform a range of functions in physiological and pathophysiological conditions of the CNS. This review considers the function and immunomodulatory mechanisms of C1q; the emerging role of C1q on synaptic pruning in developing, aging, or pathological CNS; the relevance of C1q; the complement pathway to neurodegenerative diseases; and, finally, it summarizes the foods with beneficial effects in neurodegenerative diseases via C1q and complement pathway and highlights the need for further research to clarify these roles. This paper aims to provide references for the subsequent study of food functions related to C1q, complement, neurodegenerative diseases, and human health.

## 1. Introduction

In 1894, Jules Bordet discovered the presence of a thermally unstable component or group of components of serum that acted as a complement to antibodies and interacted with bacteria. Based on its function as an antibody supplement, Paul Ehrlich named the new substance “complement” (Figure 1a). Complement is a multimolecular self-assembling biologically active system consisting of nearly 20 plasma proteins with different physicochemical and immunological properties. When activated, the precursor molecules react in a chained enzymatic reaction in a particular order. In this process, C1q acts as a promoter of the complement pathway and participates in the body’s defense by binding to antigen–antibody complexes to activate the classical complement pathway. However, in addition to the traditional role of complement activation, C1q plays other important roles in a complement-independent manner. Recent investigations have unveiled that the complement molecule C1q is involved in both neuroprotection and neurodegenerative diseases. It has been shown that C1q levels in the brains of Alzheimer’s disease (AD) mouse models are significantly higher than in normal rats [1]. C1q plays a role in neurodegenerative diseases by inducing microglia to phagocytose synapses, which ultimately results in cognitive–behavioral deficits caused by synaptic loss. The inhibition of C1q activity resulted in a diminution of microglial count, a mitigation of initial synaptic attrition, and a notable amelioration in cognitive memory impairments [2]. Suggesting that the induction of microglia phagocytosis of synapses by C1q plays an important role in neurodegenerative diseases.

Synapses are the fundamental structures of neural networks that translate information into various brain regions and cell types. Throughout the developmental journey, the establishment of fully fledged neural networks necessitates the judicious eradication of unsuitable synaptic bonds, an intricate choreography wherein microglia and astrocytes partake, orchestrating synaptic honing under the tutelage of C1q [3]. C1q is neuroprotective during the initial stages of CNS injury, where it can hold a pivotal position in avoiding the release of cell-damaging factors by enhancing the phagocytosis of microglia and macrophages and regulating inflammation [4]. However, when excessive activation of C1q occurs, it may lead to direct synaptic damage. C1q induces localized activation of microglia and astrocytes, which, by acting synergistically with other proinflammatory pathways, ultimately leads to memory-related neurodegenerative disorders such as AD, Huntington’s disease (HD), and traumatic brain injury (TBI) [5,6]. The clinical hallmark of Parkinson’s disease (PD) is a motor syndrome characterized by bradykinesia, rest tremor, and rigidity, as well as changes in posture and gait [7]. In PD cases, neurons seemed to be opsonized by C1q [8]. In contrast, blocking C1q activation using C1q knockdown or antibodies has been shown to significantly reduce synaptic phagocytosis and cognitive dysfunction [1].

With today’s rapid advances in medicine and the increasing emphasis on the prevention of diseases before they develop, there is a growing interest in utilizing bioactive substances in food to maintain health and combat chronic human diseases, in addition to the nutritional supply of food. A number of functional food factors have been shown to delay the development of neurodegenerative diseases by modulating C1q levels. In this article, we present a review of the recent advances in the role of C1q in neurodegenerative diseases, focusing on the specific mechanisms of synaptic elimination of C1q through the classical complement pathway in CNS development, aging, and disease, as well as the targeting of C1q by food components to slow down the progression of neurodegenerative diseases. The aim of this study is to provide a new theoretical basis to further explore the role of functional food factors in the development of neurodegenerative diseases and to hypothesize that C1q can be an effective target and marker for delaying the development of neurodegenerative diseases, such as AD.

## 2. C1q Structure and Biological Function

### 2.1. C1q Structure

C1, the inaugural sentinel within the precincts of the classical pathway inherent to the innate immune system, comprises the trinity of C1q, C1r, and C1s. The self-initiation of C1r, a serine protease, ensues, cascading into the enzymatic cleavage and activation of another akin serine protease, C1s. This orchestration is precipitated upon the embrace of antibodies or the surfaces of pathogens by C1q, serving as the harbinger of this sequence. Thus, a symphony of elements coalesces, culminating in the creation of a C1 complex, wherein one C1q molecule interlaces with twin counterparts of C1r and C1s. The unveiling of this assembly activates the classical complement pathway, precipitating a transformative shift in the collagenic domain [9].

The intact C1q molecule is a complex of 18 chains: 6A, 6B, and 6C, which are connected by covalent and noncovalent bonds to form AΒC-CBA triple-helical structural units [10,11,12]. Each chain of C1q has its ligand, and each region mediates different physiological functions by binding to different ligands and receptors, such as regulation of various immune cells (e.g., dendritic cells, platelets, microglia, and lymphocytes), clearance of apoptotic cells, a range of cellular processes such as differentiation, chemotaxis, aggregation, and adhesion, and neurodegenerative diseases and pathogenesis of SLE [13,14]. Interestingly, studies have observed that the B and C chains of C1q are highly conserved, and little ligands were recognized by B and C chains. Deoxy-D-ribose and heparan sulfate were recognized specifically by the globular domain of C1q involving interactions with ArgC98, ArgC111, AsnC113 and LysC129, TyrC155, TrpC190, respectively. Helena et al. reported that phosphatidylserine is one of the C1q ligands on apoptotic cells and interacts with subunit C of C1q, which unfolds during the nascent phases of apoptosis [15]. Moreover, plausible 3D models of the C1q globular domain in complex with C-reactive protein (CRP) and IgG were proposed; ArgB114, ArgB129, and ArgB163 were involved in the interaction with IgG, while LysA200, TyrB175, and LysC170 at the top of the C1q head directed toward CRP [16]. Furthermore, the A chain is the exact opposite, with a collagen region containing a major binding site for nonimmunoglobulin substances that can bind Adiponectin [17], Von Willebrand factor [18], C-reactive protein [19,20], Serum amyloid P [21], DNA [22], Aβ [22], and Heme [23]. Among them, IgM [24,25], LPS [26], GAPDH [27], Blood platelets [28], Fibronectin [29], Adiponectin [17], C-reactive protein [19,20], Serum amyloid P [21], and Aβ [22] can activate the classical complement pathway by binding C1q, whose binding sites are the cationic regions 14–26 and 76–92 of the C1q A chain, and the use of the same peptide as residues 14–26 can regulate ligand binding to C1q to activate the classical complement pathway [22]. In accordance, one may posit the conjecture that the preponderance of C1q’s functionalities seems to derive primarily from the A chain. The target ligands and versatile recognition properties of C1q are summarized in Table 1.

Some studies have also shown that the B-chain of C1q plays a crucial role in tumorigenesis binding of multiple tumors. Yamada and colleagues have laid bare that elevated C1qB expression exhibited a marked correlation with unfavorable prognostication within the context of renal cell carcinoma. In contrast, the work of Linnartz-Gerlach and collaborators uncovered the downregulation of C1qB in the cerebral milieu of triggering receptors expressed on myeloid cell-2 (TREM2) knock-out mice. Intriguingly, it merits noting that TREM2 has been brought to the forefront as a conduit for intracellular messaging, effectuated via its partnered transmembrane adapter, TYROBP [30]. Demonstrations have borne witness to a robust linkage existing between TYROBP and C1qB expression amidst individuals afflicted by gastric carcinoma [31]. Analogously, investigations have illuminated that a disruption within this signaling conduit engenders an array of perturbations in physiological equilibrium and confers susceptibility to an array of maladies encompassing the likes of senescence and age-linked neuronal attrition [32,33]. Concurrently, evidence has surfaced elucidating that C1q within the realm of IgG-mediated acquired immunity underscores the centrality of Arg114 and Arg129 within the B chain, marking them as pivotal residues in the tapestry of IgG binding. This discourse further accentuates the role of Arg162 in the A chain, along with Arg163 in the B chain and Arg156 in the C chain, as integral participants in the orchestration of C1q-IgG interplay [34,35]. Although there are fewer studies on the C1qC chain, it is clear that each of the three C1q chains may be functionally independent and capable of differentially splicing ligands.

The globular “head” of each AΒC chain is connected to a fibrillar central region by six collagen-like “stalks” that form two distinct structural and functional domains: the collagen-like region (cC1q) and a spherical “head” structure (gC1q) [36]. The ultimate configuration of the C1q molecule mirrors the semblance of a cluster of tulip blossoms (Figure 1b). Interestingly, these two domains acted as separate parts from each other and interacted with diverse biological structures, including pathogen- and cell-associated molecules. The collagen-like regions of C1q are engaged in immune response effector mechanisms through their interaction with a tetramer of complement C1r and C1s proteases (C1r2s2) or receptors on immune cell surfaces [37,38]. More prominently, the globular regions support the recognition properties of C1q with the striking ability to sense a wide variety of ligands [39,40]. (As shown in Table 1) In general, gC1q was performed to investigate the location binding interactions and recognition specificity of C1q-ligands complexes, as well as in the regulation of C1 activation, such as the lipopolysaccharides (LPS) inside Table 1. LPS interacts specifically with the gC1q domain in a calcium-dependent manner. LPS and IgG-binding sites on the gC1q domain appear to be overlapping, and this interaction can be inhibited by a synthetic C1q inhibitor, suggesting common interacting mechanisms [26]. However, it does not work when only the C1q tail or C1q globules are present; in other words, the complete C1q is required to affect the organism [4].
foods-12-03580-t001_Table 1Table 1The versatile recognition properties of C1q.C1q LigandC1q Binding Region/Binding SiteFunctionRef.IgMC1q globular domainActivate the classical complement pathway[24,25]IgGC1q globular domainActivate the classical complement pathway[16]LPSC1q globular domainActivate the classical complement pathway[26]GAPDHC1q globular domainActivate the classical complement pathway[27]Blood plateletsC1q globular domainActivate the classical complement pathway[28]CRPC1q globular domainActivate the classical complement pathway[5,19,20]Pentraxin 3C1q globular domainInteracts with C1q and inhibits the classical complement pathway[41]FibronectinC1q globular domainActivate the classical complement pathway[29]CalreticulinC1q globular domainrecognize apoptotic cells[42,43]HeparinC1q globular domainInhibit the classical complement pathway[44]ApoEC1q stalkInhibit the classical complement pathway[45]AdiponectinGlobular domain of the C1q A chainActivate the classical complement pathway[17]Von Willebrand factorN-terminal of the C1q A chainInhibit the classical complement pathway[18]Serum amyloid PResidues 14–26 and 76–92 of the C1q A chainActivate the classical complement pathway[21]DNAResidues 14–26 of the C1q A chainActivate the classical complement pathway[22]AβResidues 14–26 of the C1q A chainActivate the classical complement pathway[22]HemeTyrA122 of the C1q A chainInhibit the classical complement pathway[23]Deoxy-D-riboseResidues Arg98, Arg111, Asn113 of the C1q C chainInhibit C1 activation[44]Heparan sulfateResidues Lys129, Tyr155, Trp190 of the C1q C chainInhibit C1 activation[44]PSGlobular domain of the C1q C chainEfficient apoptotic cell removal determined synaptic vulnerability[15,46,47]

### 2.2. C1q Biological Functions

C1q serves as the herald of the complement pathway, adroitly bridging the realms of innate and adaptive immunity. In the context of innate defense, the dearth of C1q is intimately tied to the constellation of systemic lupus erythematosus (SLE) [48]. Within this intricate framework, one of the conjectured mechanisms, aptly denominated the waste disposal paradigm [49], propounds that C1q assumes an indispensable role in the expunction of deceased cells. In the unfortunate absence of C1q, this regulatory function falters, impeding the efficient clearance of expired cellular constituents [50]. The cascading result is the exposure of intracellular moieties, inciting an immune retort against these very (self) entities [51]. In addition to its influence within the realm of innate immunity, C1q unfurls a diverse spectrum of roles within adaptive immunity. Human T cells bear the imprint of C1q receptors, bestowing upon C1q the capacity to navigate the complex terrain of T-cell responses. Recent investigations in mouse models have cast light upon the intriguing prospect that C1q might indeed wield a discernible influence on T-cell immunity. Within mouse models emblematic of SLE, a striking phenomenon emerges: C1q assumes the role of an inhibitor, quelling the vigor of CD8 T-cell responses [52]. The authors propose a mechanism in which C1q is internalized and impacts mitochondrial function via C1q receptors. In emphysema, the absence of C1q leads to a shift from suppressive regulatory T-cell responses to pro-inflammatory Th17 responses [53]. The synergy between human observational studies and mouse models showed that C1q acts on the T-cell suppressor [3]. From these studies, it is clear that C1q stands as a regulator, dictating the amplitude and caliber of T-cell responses, reflecting the important role of C1q in adaptive immunity. In contradistinction to the diminished levels of C1q noted across various autoimmune conditions, an augmentation in C1q levels has been documented within the annals of numerous infectious and neurodegenerative maladies, including AD and PD. Hence, increasing awareness of the distinct roles of C1q, as well as the relationship between C1q level with autoimmune diseases and neurodegenerative diseases, highlights the need for a thorough understanding of C1q in innate immunity and adaptive immunity.

## 3. Complement in the Brain

The complement cascade is a key effector mechanism of the innate immune system that contributes to the rapid clearance of pathogens and dead or dying cells and increases the extent and limits of the inflammatory immune response. C1q, the promoter of the classical complement pathway in the complement cascade, has been clearly shown to play a beneficial role in synaptic elimination during neurological development, but excessive C1q-mediated synaptic pruning in the adult or injured brain may be detrimental in a variety of neurodegenerative diseases.

### 3.1. C1q and the Complement Pathway

The complement pathway stands as a crucial constituent nestled within the precincts of the innate immune system that can participate in host defense by rapidly recognizing and eliminating pathogens, cellular debris, and misfolded proteins, facilitating the clearance of dead cells or antibody–antigen complexes [54]. In response to multiple endogenous or exogenous substances, the components of the complement body are activated sequentially in a chain reaction to produce various biological effects.

The classical complement pathway, lectin pathway, and alternate pathway are three commonly accepted pathways of complement system activation. The complement pathways share commonalities but also have their characteristics. The three complement pathways are mainly initiated by promoters, mediated by C3 converting enzyme and C5 converting enzyme, and activated by a series of pathway amplification reactions to activate the complement system and form membrane attack complex (MAC), resulting in the lysis and rupture of the plasma membrane of the antigen-bearing cells, and the outflow of the cellular contents, which triggers the inflammatory response [55]. The details are shown in Figure 2.

Similarly, the lectin pathway’s activation mirrors that of the classical pathway, albeit with a distinct initiator—mannose-binding lectin (MBL). MBL undertakes the task of forming multimeric lectin complexes upon engaging ficolin. This intricate association ushers in the activation of MBL-associated serine protease (MASP), heralding the initiation of the complement pathway. MASP-1 and MASP-2 bear semblance to C1r and C1s, respectively. Their counterparts in this realm, MASP-1 and MASP-2, wield the power to usher in the complete activation of the complement system by cleaving C4 and C2, thus assembling the C3 convertase.

Diverging from the trajectory of classical activation, the alternative pathway circumvents the involvement of C1, C4, and C2. Instead, it forges a direct path towards the activation of C3, thereby igniting a cascading sequence that propels the components from C5 through C9 to fulfillment. Meanwhile, the lectin pathway and the classical pathway formation of C3 convertase from C4b and 2a, whereas the alternative pathway is via C3b and Bb. The alternative pathway can play a significant anti-inflammatory role as a low-level sustained activation of nonspecific immunity by acting directly on invading microorganisms and other foreign bodies at the early stage of pathogen infection before specific antibodies are produced.

Given that the initiation of the classical complement pathway hinges upon the genesis of antigen–antibody complexes before their binding with C1q, and the birth of antibodies necessitates a specific immune retort, it follows that the classical complement pathway predominantly exerts its influence during the latter phases of infection.

### 3.2. C1q through Classical Pathway Activation Mediated Synapse Pruning

Within the complement pathways, it is the classical complement pathway that predominantly takes on a significant role within the CNS, in which C1q acts as a recognition protein and plays different roles at different stages of the CNS.

#### 3.2.1. C1q in Developing CNS

During the development of the CNS, precise neural circuits are vital for the development and functional maturation of the brain. Complement pathway activation products are crucial to modulate synapse formation to avoid redundant, spider web-like neural network growth. A subset of synapses in neurons is pruned while other presynaptic or postsynaptic axons from the parent neuron are positively stabilized and strengthened [56]. C1q activates the downstream complement component C3 by labeling abnormal synapses or inappropriate synaptic connections, and C3 binds to the receptor CR3 on microglia, initiating the complement pathway and the exercise of synaptic pruning by microglia (Figure 3). In this way, it participates in beneficial synaptic elimination and promotes neuronal development and functional maturation.

Synaptic pruning removes less active or “weak” synapses, strengthens synaptic connections appropriately, and promotes neuronal maturation, meaning that C1q can “detect” morphofunctional changes in synapses [57]. As long as the “weak” synapses are labeled by C1q and subsequently removed by phagocytic microglia via synaptic pruning [55,58]. To what kind of synapses are “weak” that contribute to labeling by C1q during the synaptic pruning remains unclear.

Györffy et al. divulged the existence of processes akin to apoptosis within synapses adorned with C1q, as unveiled through a proteomic exploration and pathway analysis of synaptosomes labeled with C1q [58]. The presence of apoptosis-like processes in C1q-labeled synapses and the fact that synaptic apoptosis induces increased C1q levels suggest that synaptic pruning based on C1q labeling may be triggered by synaptic apoptosis, which can be inhibited by increased synaptic activity. In conditions of normalcy, the proclivity for C1q to bind surfaces leans toward late apoptotic cells in comparison to their early apoptotic counterparts [59]. It has also been shown that microglia preferentially phagocytose apoptotic cells, whereas in the absence of C1q, the level of phagocytosis by microglia is reduced [60]. Bioinformatics analysis showed that C1q-labeled synapses contained higher levels of caspase 3 and caspase 5, both markers of apoptosis, compared with C1q-negative synapses [58], suggesting that synaptic pruning and clearance of apoptotic cells are similar in the mechanism. Phosphatidylserine (PS) is one of the C1q ligands on apoptotic cells [15]. An apoptosis-like process exists in C1q-labeled synapses, and the synaptic apoptotic mechanism induces elevated C1q levels, while synaptic pruning based on C1q labeling may be triggered by gC1q-PS binding effects [44]. During normal growth and development, synapses may be induced by increased apoptotic signaling while C1q levels are increased and aggregated toward them, while C1q stimulation also causes apoptotic cells and cell debris clearance. In contrast, in neurodegenerative diseases, synapses may be triggered to apoptosis by increased levels of C1q, leading to their “misrecognition” by microglia and their engulfment.

The perturbation of C1q’s equilibrium precipitates anomalies in synaptic pruning, culminating in compromised synaptic transmission, diminished cerebral connectivity, impairments in social engagement, and heightened instances of repetitive behavioral traits. This intricate web of repercussions interweaves with the realms of epilepsy, as well as a spectrum of neurodevelopmental and neuropsychiatric afflictions [61]. Concurrently, mice lacking in the complement protein C1q or its downstream counterpart, complement protein C3, manifest pronounced and enduring deficiencies in the process of CNS synapse elimination. This is starkly evidenced through the thwarting of anatomical refinement within retinogeniculate connections, along with the retention of superfluous retinal innervation amongst lateral geniculate neurons [62]. In a study involving a mouse model of the CNS, C1q was upregulated after birth and peaked within two weeks [3]. Increased synaptic connectivity and epileptic activity were shown in C1q knockout mice that were unable to establish proper synaptic connections [63].

#### 3.2.2. C1q in Aging

The wane of cognitive prowess has surfaced as a formidable health menace during the later stages of life. Within this purview, the decline in cognitive function is intrinsically linked to the perturbed neuronal circuitry that comes with advancing age. Unlike the rise in C1q levels during development, neuronal C1q is normally downregulated in the adult CNS. Certain inquiries have unearthed a notable resurgence in C1q protein levels within the confines of the aging mouse and human brain [64,65]. In harmony with the rise in serum C1q concentrations that accompany the passage of time, a commensurate augmentation in C1q levels within the cerebrospinal fluid (CSF) has been duly noted [66]. Significant disparages in the C1q protein abundance within mouse and human brain tissue emerged palpably through the lens of immunohistochemistry, particularly when juxtaposing brain specimens from the early stages of life to those in advanced age [67]. This escalation predominantly manifested near synapses, materializing at the nascent stages and with remarkable intensity in specific cerebral domains. Among these locales, a subset—but not the entirety—comprising regions acknowledged for their predisposition to neurodegenerative disorders underwent this surge, i.e., the hippocampus, substantia nigra, and piriform cortex [67]. C1q-deficient mice exhibited enhanced synaptic plasticity in the aging and reorganization of the circuitry in the aging hippocampal dentate gyrus [10].

#### 3.2.3. C1q in Diseases

Changes in the expression level of C1q in different diseases have different effects on the disease process or pathological changes. Diminished levels of C1q have surfaced within the context of autoimmune disorders, select inflammatory ailments, and numerous tumor types. Notably, an elevation in anti-C1q antibodies has been unveiled to precede renal flares in lupus. Furthermore, autoantibodies targeting C1q have also been documented in the annals of diverse spontaneous mouse models of systemic lupus erythematosus (SLE) [68]. C1q deficiencies have been correlated with heightened vulnerability to infections triggered by encapsulated bacteria, particularly cases of pneumonia and meningitis. Additionally, recurrent respiratory infections are known to afflict individuals with such deficiencies [69]. The conceptualization of C1q’s role in the genesis of cancer remains in a state of continuous evolution. Evidence has emerged indicating that elevated C1q levels bear a favorable prognostic implication for disease-free survival in basal-like breast cancer, as well as for overall survival in HER2-positive breast cancer. However, a converse narrative unfolds when exploring its influence within the landscape of lung adenocarcinoma and clear cell renal cell carcinoma, where C1q appears to assume a pro-tumorigenic role [70]. C1q seems to encompass a binary function within the realm of cancer, manifesting itself as both a catalyst for tumor advancement and a guardian against tumors, its stance oscillating in tandem with the intricate context of the disease at hand. Elevated C1q levels have been detected across a spectrum of domains, including aging, infectious diseases, assorted inflammatory disorders, and cardiac and cerebral ailments. In the context of infectious diseases, the surge in circulating C1q levels is likely attributed to an escalated production of C1q as a countermeasure against pathogens. However, the localized surges noted in conditions such as neuroborreliosis and meningitis could potentially stem from augmented leakage from the circulation. C1q can regulate cytokine expression, and when cells are injured or encounter trauma, the concentration of free C1q becomes high, which can stimulate neutrophils to move toward the lesion [71]. Also, neutrophils are recruited to the site of cell injury accompanied by the accumulation of C1q, IgM, and albumin; among others, accumulation of C1q, IgM, and albumin is accompanied by the accumulation of these proteins [4]. Also, in neurodegenerative diseases, increased C1q levels are a consequence of increased (local) C1q production. C1q engages in a deleterious interaction with anomalous protein aggregates, thus embroiling itself in the genesis of neurodegenerative afflictions like AD alongside other neuropsychiatric disorders [71]. Mouse models lacking C1q, although showing the “plaque” structure characteristic of AD, those mice lacking C1q have significantly less inflammation around these plaques and significantly more neuronal integrity compared with transgenic mice with an intact complement system [72]. Studies have shown that ANX005 is a potent anti-C1q-targeting antibody that binds to C1q, inhibits its interaction with multiple substrates, and prevents classical complement pathway activation while leaving the lectin and alternative complement pathways intact [73].

### 3.3. The Role of C1q in the Complement-Independent Manner

Indeed, C1q is widely acknowledged for its role as an initiator within the classical complement pathway, culminating in the activation of the complement system. This activation intricately aids in the swift elimination of pathogens and dead cells. It is of significance to note that C1q assumes roles that extend beyond its involvement in the complement pathway, encompassing the maintenance of homeostasis and regulatory functions in a manner distinct from its engagement within the complement system. There is some evidence that C1q can be synthesized in peripheral tissue bone marrow cells in the absence of C1q-associated C1 serine proteases C1r and C1s [74]. Within the CNS, the synthesis of C1q experiences an augmentation as an initial retort to injury in numerous instances, yet this response occurs independently from the simultaneous synthesis of the C1 serine proteases, indicating C1q-mediated activities individually without a complement pathway. Demonstrations have illuminated that C1q possesses the ability to induce a gene expression program that champions neuroprotection. In doing so, it potentially furnishes a shield against the menace posed by Aβ, particularly in the preclinical phase of AD and other neurodegenerative pathways [75]. Moreover, C1q presents a protective impact on primary neuronal viability in rodents from Aβ and serum amyloid P (SAP)-induced neurotoxicity in the absence of other downstream factors of the complement pathway [4]. C1q, distinct from C1, engages with myeloid cells, which encompass microglia, fostering an expedited elimination of apoptotic cells and neuronal fragments, thereby curbing the production of pro-inflammatory cytokines [76]. Another study has proved that C1q, heightened in vivo as an early rejoinder to injury, disengaged from the simultaneous upregulation of other complement constituents, has the potential to incite a genetic orchestration fostering neuroprotection [75]. In the presence of blood–brain barrier (BBB) dysfunction, C1q was accompanied by a rapid increase in proinflammatory factors such as interleukin-1β and tumor necrosis factor α but not in C1q-associated C1 serine proteases C1r and C1s [70]. This suggests that C1q can exert neuroprotective effects independently of the classical complement pathway. These results reveal a role for C1q in physiological changes and pathological changes without activating the classical complement pathway.

## 4. Local Synthesis of C1q in the Brain

Initially, the presence of C1q was studied as a component of the immune system in the blood, with its main site of synthesis in the liver. However, it is now recognized that under conditions of inflammation, external injury, and cellular stress in the CNS, C1q can be expressed and regulated from a variety of cell types, including neurons, microglia, and astrocytes.18 Each of these cells plays a role in regulation, homeostasis, and destruction in the CNS.

### 4.1. C1q and Microglia

Microglia are the resident immune cells of the brain, and their dysfunction may contribute to neurodegenerative and psychiatric disorders [77]. C1q is synthesized within the cerebral domain, primarily by microglia. In states of equilibrium, C1q is upheld at a subdued magnitude. Yet, when microglia are galvanized into action, the amplitude of C1q escalates, heralding the stimulation of pro-inflammatory cytokines such as interleukin-6 (IL-6) and TNF-α, thereby culminating in the demise of neurons [70]. Simultaneously, the affinity of C1q for apoptotic cells or neuronal fragments sets forth an augmentation in phagocytic activity within microglia. Remarkably, there are instances when heightened C1q levels yield a transient surge in reactive oxygen species (ROS), nitric oxide (NO), and calcium. Furthermore, this surge in C1q can serve to arrest the proliferation of microglia [78].

Microglia are activated in response to stimulation and accompanied by transcriptional adaptive functional changes, and microglia can generally be differentiated into two extreme states: the classical (M1) phenotype and the alternative (M2) phenotype [79]. M1-type microglia release proinflammatory factors and toxic substances, which enhance brain injury. M2 microglia, on the other hand, achieve neuroprotective effects by promoting tissue repair and regeneration [80]. Microglia are the main source of C1q in the brain [81]. This elucidates the rationale behind the presence of C1q not only within the circulatory system but also within the CNS. In states of equilibrium, the levels of C1q are deliberately maintained at a subdued magnitude; however, when microglia are overactivated, C1q levels increase rapidly [78]. M1-type microglia can play a protective role in the early stages of neurodegenerative diseases by secreting C1q involved in the clearance of protein aggregates. However, as the disease progresses, the deleterious effects of M1-type microglia outweigh their beneficial effects, and their clearance becomes less efficient. Moreover, neuronal dysfunction, injury, and degeneration result from the release of large amounts of C1q [82,83].

### 4.2. C1q and Astrocytes

Astrocytes are among the most numerous cells in the CNS and are critical for potassium homeostasis, neurotransmitter uptake, synapse formation, BBB regulation, and nervous system development [84]. Recent studies have found that neuroinflammation and ischemia induce two distinct phenotypes of astrocytes, control microglia “M1” and “M2” named “A1” and “A2”. The A1 phenotype is close to that induced by lipid polysaccharide-induced neuroinflammation, acute CNS injury, and the underlying pathology of many neurodegenerative diseases [85]. In contrast, A2-type astrocytes strongly promote neuronal survival and repair. Moreover, activated microglia can induce neuronal death by secreting C1q to induce astrocytes to shift to the A1 phenotype [18]. In contrast, genes expressed by A1-type astrocytes were significantly downregulated in the absence of C1q, suggesting an important role of C1q in the polarization of astrocytes toward the A1 phenotype [86]. It has been shown that *Astragalus* polysaccharide inhibits the formation of A1-type astrocytes by inhibiting C1q secretion by microglia [87]. In patients with multiple sclerosis, the detection of CNS plaques is an important marker for the co-localization of C1q with reactive astrocytes [88]. A slew of contemporary investigations have unveiled the prospect that C1q originating from astrocytes might exert influence over the attenuation and deterioration of neuronal synapses amidst the trajectory of neurodegeneration [89]. The augmented expression of reactive astrocyte genes spurred by aging was notably dampened in mice devoid of microglial-secreted C1q, a stimulator recognized for inciting the formation of A1 reactive astrocytes. This observation signifies that microglia wield a role in fomenting astrocyte activation during the aging process. Simultaneously, A1 reactive astrocytes forfeit their capacity to execute customary functions. Moreover, the escalated up-regulation of reactive genes within astrocytes triggered by aging might underpin the cognitive regression witnessed within susceptible cerebral domains during normal aging. Additionally, this accentuates the heightened vulnerability of the aged brain to damage [86]. Chen and colleagues’ work showcased the inhibitory prowess of Cholecystokinin octapeptide in curtailing the induction of A1-reactive astrocytes by diminishing C1q levels. This intervention concurrently stimulated the genesis of glutamatergic synapses, thus fostering neurocognitive resurgence within aged dNCR model mice [90].

### 4.3. C1q and Neurons

Neurons integrate thousands of synapses to process and transmit information [3,91]. Whereas neurons are another cell regulating C1q secretion in the brain in addition to astrocytes and microglia, neurons respond to neuroinflammation and other inflammatory mediators [92,93]. The complement factor C1q, sourced from cells within the central CNS, is also intrinsically linked to conferring neuroprotection against external infections [94]. C1q is deemed advantageous in the elimination of aggregated proteins after the activation of the complement factor due to the engagement of low levels of aggregates. C1q production is induced in large numbers when neurons are damaged, enhancing neuronal activity and protecting neurons from Aβ- and SAP-induced neurotoxicity [4]. Nonetheless, when the complement factor is persistently triggered, it can prove detrimental to the CNS as a result of the activation of microglia and the subsequent release of pro-inflammatory cytokines [95]. Interestingly, cholesterol distribution and levels are also influenced by C1q in neurons, which can enhance neurons indirectly by regulating cholesterol levels, and C1q affects neuronal construction by regulating lipid metabolism and membrane-associated gene expression [96].

C1q plays different roles under different cellular expressions and regulations and ultimately serves different roles in a variety of neurodegenerative diseases. In the subsequent overview, we use C1q as an entry point to systematically understand the various mechanisms of neurodegenerative diseases.

## 5. C1q in Neurodegenerative Diseases

Beyond its established roles in CNS growth, development, and bodily immunization, a novel facet of C1q’s functionality has been freshly unveiled within the intricate tapestry of neuropathological pathways that underpin neurodegenerative disorders and TBI. This revelation has cast a spotlight on C1q as a prospective therapeutic avenue for safeguarding neuronal well-being or for retarding the progression of neurodegenerative maladies.

### 5.1. Alzheimer’s Disease

In a series of experiments, excessive complement-mediated synapse pruning was found to be involved in the process of forgetting in AD. Region-specific loss of synaptic salience is a more potent contributor to cognitive decline in AD than the hallmark features of AD, Aβ plaques, and Tau protein hyperphosphorylation [1]. Unlike the state of synapses requiring proper pruning during growth and development, it has been shown that the number of synapses is significantly reduced in patients with early AD. The number of synapses in 75% of patients with mild cognitive impairment was lower than the average of normal individuals, and the number of synapses correlated significantly with the cognitive–behavioral status of AD patients [97,98,99]. Similarly, synapses in the temporal cortex are reduced by 38% and in the frontal cortex by 14% in AD patients compared with normal subjects [98]. AD mice lacking C1q display reduced synapse loss, supporting a role for C1q in mediating synapse removal [2]. At the end of AD, synapse number decreases positively with the degree of cognitive–behavioral impairment in AD patients [99]. Interestingly, C1q can be observed colocalized with either pre-synaptic or post-synaptic markers in animal models of aging-related diseases, including AD. Accordingly, the upregulation of C1q-tagged synapses is also proved in AD and other neurogenerative disorders-induced cognitive loss. Conversely, the knockdown or blockade of C1q in mouse models of AD has been shown to protect synapses and prevent cognitive impairment, suggesting the detrimental influence of C1q in synaptic loss, and even C1q-labeled synaptic loss may directly contribute to the worsening of AD.

AD is characterized by synaptic dysfunction and neurodegeneration, which are often caused by the deposition of Aβ plaques and neurofibrillary tangles [100]. The deposition of Aβ plaques triggers a series of chain reactions that lead to intracellular Tau protein misfolding and assembly, which subsequently allows the spread of the lesion throughout the neural circuit as well as the cortex, ultimately leading to neurological failure and cognitive decline. C1q plays an important role in this. It has been shown that blocking C1q activation by genetic or antibody-mediated means can attenuate the toxic effects of Aβ and hyperphosphorylated Tau on synapses [101]. This provides another direct evidence for a deleterious effect of C1q during the process of AD (Figure 4).

#### 5.1.1. C1q and Aβ in AD

The major component of amyloid plaques is Aβ, a peptide with 39 to 43 amino acids derived from amyloid b protein precursor (APP) [102]. Studies have shown that the imbalance between the production and clearance of Aβ and related Aβ peptides plays a fundamental role in the pathogenesis of AD [103,104]. In vitro experiments have shown that C1q interacts with Aβ through its A-chain residues 14–26 [104,105,106,107]. The complement component C1q has nearly 100% co-localization of Aβ in humans with AD and in mouse models of AD [108]. Sections of the Aβ-treated hippocampus showed a significant increase of C1q in the hippocampus [109]. When soluble Aβ oligomers were injected into the lateral ventricles of WT mice, Aβ oligomers were found to induce C1q deposition [1]. Similarly, when J20 mice were injected with a γ-secretase inhibitor that rapidly reduced Aβ production, it significantly reduced soluble Aβ levels in mice with a corresponding reduction in C1q deposition. When C1q knockdown was followed by Aβ injection, the synaptic loss induced by Aβ was significantly reduced [2]. The use of anti-C1q antibodies similarly prevented Aβ-induced synaptic loss in mice [1], suggesting that C1q is required for Aβ-induced synaptic loss in vivo. Notably, C1q knockdown does not affect Aβ deposition [110], so C1q may function downstream of Aβ. Aβ appears abnormally as early as 20 years before the onset of overt clinical symptoms [111]. Aβ deposition is the beginning of neurodegenerative lesions, but the accumulation of hyperphosphorylated Tau proteins is the main driver of the deteriorating pathological process.

#### 5.1.2. C1q and Tau in AD

In AD and other Tau lesions, Tau aggregates in an abnormally phosphorylated form in the torso region of neurons and can localize to synapses, where it disrupts synaptic plasticity and leads to synaptic loss [112]. Positron emission tomography (PET) imaging targeting Aβ and Tau has unveiled a consequential relationship: the velocity of amyloid aggregation forecasts the advent of Tau accumulation, which in turn heralds the initiation of cognitive decline [113]. In AD, Tau protein aggregation may begin in the entorhinal cortex and then propagate to the hippocampus, as well as within the limbic cortex, reflecting the progression of AD patients from asymptomatic, mildly symptomatic, to full dementia [114]. It has been shown that hyperphosphorylated Tau protein induces more C1q aggregation at the synapse than Aβ plaques [101]. In the mouse model, increased Tau phosphorylation and accumulation were accompanied by a dose-dependent increase in C1q [115]. It has been shown that knockout of the granule protein precursor gene PGRN significantly reduces Aβ plaque production, but deletion of PGRN enhances C1q deposition at the synapse while increasing the accumulation of hyperphosphorylated Tau protein in the hippocampus [116]. Interestingly, knockdown of the transmembrane immune signaling adaptor TYROBP showed opposite results, where knockdown of TYROBP resulted in a significant reduction in C1q and improvement in memory cognition impairment, hyperphosphorylated Tau protein was not reduced by the decrease in C1q, but instead, its spread was further expanded and Tau protein phosphorylation levels were increased. Suggesting that when there are multiple competing effects occurring simultaneously, the deleterious effects of increased Tau protein phosphorylation levels and diffusion can be overcome as long as the beneficial effect of a significant decrease in C1q is large enough [117]. It may give insights into the target role of C1q in regulating the progression of AD pathology and cognitive loss.

### 5.2. Parkinson Disease

PD is clinically characterized by an akinetic rigid syndrome related to reduced. This syndrome is entwined with a decline in dopamine levels within the striatum, arising from the gradual demise of terminals belonging to degenerating neuromelanin-containing dopaminergic neurons in the substantia nigra pars compacta [118]. A handful of investigations have scrutinized the complement system within the context of the PD brain. The steady manifestation of C1q was discernible only within microglial cells spanning the cerebral expanse. After MPTP (1-methyl-4-phenyl-1,2,3,6-tetrahydropyridine) exposure, there was an early and temporary elevation in microglial C1q expression within the substantia nigra and striatum, as unveiled through techniques of immunohistochemistry and in situ hybridization. Notably, Rozemuller and collaborators found no immunostaining indicative of C1q within cortical Lewy bodies [119]. Concurrently, mice devoid of the C1q protein exhibited no substantial differences in terms of the loss of nigral dopaminergic neurons, striatal dopaminergic fibers, and dopamine levels induced by MPTP in comparison to their control counterparts [120,121]. This shows that C1q is not a major contributor to cognitive impairment in PD. Simultaneously, within the substantia nigra pars compacta (SNc) of PD cases, there manifested an augmented accumulation of extracellular neuromelanin within the tissue, a manifestation arising from the degeneration of dopaminergic neurons. In this milieu, neuromelanin granules and fragments from deteriorated neurons appeared to be tagged by C1q, thus becoming subject to phagocytosis by C1q-positive microglia and macrophages situated both within the tissue and around perivascular spaces. Notably, cells bearing neuromelanin and C1q also adhered to the inner surfaces of blood vessels in the SNc in the context of PD [8]. Hence, microglia demonstrate the capability to engulf and eliminate cellular detritus emanating from degenerating neurons within the SNc, effectively orchestrating this process through a pathway facilitated by C1q, a phenomenon that occurs within the context of PD. Although C1q may not play a direct pathological role in PD, it can affect the disease process through microglia phagocytosis, etc. Therefore, when we focus on the role of C1q in PD, we should not only look at its expression but also pay attention to the related pathways or effects on other cells.

### 5.3. Huntington’s Disease

HD is an autosomal-dominant neurodegenerative disorder characterized by a relentless progression, culminating in targeted neuronal attrition and impairment, predominantly within the striatal and cortical regions [122]. Within the striatal milieu of HD, a convergence was observed wherein neurons, myelin, and astrocytes demonstrated a spatial overlap with C1q. In contrast, no C1q was found in the normal striatum. In normal control brains, the abundance of C1q mRNA ranged from 2 to 5 times lower when juxtaposed with the levels identified in the striatum affected by HD. The course of HD is marked by a neuroinflammatory progression orchestrated by the activation of microglia within the cerebral domain [123]. Astrogliosis and microgliosis were apparent in all caudate and internal capsule samples from individuals with HD, a phenomenon absent in normal brain tissue. Microglia of the M1 phenotype within the HD context produced C1q, which was subsequently triggered on neuronal membranes. This dual action of C1q not only facilitated neuronal necrosis but also contributed to proinflammatory activities [124]. Meanwhile, it has been reported that the secretion of cytokines C1q upon M1 microglial activation can induce the generation of reactive A1 astrocytes at neuronal structures, which play a major role in brain motor coordination [125,126]. These intricate processes are believed to precipitate neurodegenerative events within the brain, ultimately giving rise to the motor dysfunctions that become manifest in the later stages of this neurological affliction. Kaempferol, a natural antioxidant found in vegetables and fruits consumed as part of human nutrition, has exhibited the capacity to forestall the proteolytic activation of complement C1q protein and the subsequent emergence of reactive A1 astrocytes. This phenomenon has been observed in the context of 3-nitro propionic acid-induced injury within the striatum and hippocampus. Cognitive–behavioral deficits in experimental animals significantly improved when microglia secretion of C1q was reduced in an animal model of HD [127].

### 5.4. Traumatic Brain Injury

TBI emerges as the most potent environmental catalyst in the emergence of AD and other neurodegenerative disorders linked to dementia. The initial trauma sustained by the brain impairs the integrity of the BBB, consequently permitting the infiltration of peripheral circulating macrophages into the cerebral milieu. This occurrence subsequently accentuates the inflammatory response [128]. The prevailing notion suggests that a transition between the M1 and M2 microglial phenotypes transpires within the framework of TBI. However, it appears that there exists a proclivity towards favoring the M1 phenotype over the M2 phenotype in the context of TBI-associated secondary injury [129]. In pathological scenarios encompassing TBI, there is substantiated evidence suggesting that C1q plays a contributory role in steering a shift toward the M1 phenotype [78]. In tandem with the direct harm incited by M1 microglia, the C1q they release can also instigate the activation of astrocytes [130]. A prominent constituent of the inflammatory pathway, the complement system, often escapes notice, yet it too undergoes activation as an integral facet of the neuroinflammatory rejoinder in TBI [131]. The intrinsic complement system within the CNS undergoes activation, with this activation further amplified by an influx of complement components from the bloodstream, facilitated by the disruption of the BBB. In parallel, certain investigations have demonstrated a noteworthy accumulation of C1q on synapses located in the hippocampus. This accumulation aligns temporally with the loss of synapses 30 days after the injury. Significantly, both genetic interventions and the implementation of pharmacological measures to obstruct the complement pathway yielded the prevention of memory deficits in aged animals subjected to injury [132,133]. Therefore, strategically focusing on the modulation of C1q emerges as a substantial avenue for potential clinical intervention after TBI within the aging demographic (Table 2). 

## 6. Efficacy of Dietary Food Related to C1q for Memory Improvement

As one of the most important mechanisms of CNS, C1q plays an important role in the regulation of brain environment balance via the classical complement pathway, as discussed above. Recently, a body of literature has demonstrated that numerous food compositions exert neuroprotective effects via regulating C1q, providing evidence for the CNS of food related to C1q. Several common dietary food components related to the regulation of C1q for CNS health are shown in Figure 5.

*Artemisia annua* L., a herbaceous plant with heat-clearing properties, has garnered renown due to its antimalarial compound, artemisinin [135]. Moreover, it has garnered heightened interest owing to its demonstrated anti-inflammatory and immunoregulatory capabilities. Intriguingly, the acidic homogeneous polysaccharides derived from *Artemisia annua* have exhibited noteworthy efficacy in anticomplement activities through the classical pathway and alternative pathway. *Prunella vulgaris*, a perennial plant with a broad geographical distribution encompassing China, Japan, and Europe, has been employed for its spikes in a pivotal role within an herbal tea cherished in southern China for its ability to dissipate pathogenic heat from the bloodstream [136]. Interestingly, homogeneous acidic polysaccharides extracted from the spikes of *Prunella vulgaris* exhibit a capacity to interact with C1q, exerting an influence on the C2, C3, C5, and C9 constituents of the complement system. This property renders it potentially valuable in addressing ailments correlated with the excessive activation of the complement system. *Viola tianshanica Maxim*, a perennial herbaceous plant, finds its distribution primarily in Central Asia, notably within the Xinjiang Uygur Autonomous Region of China [137]. An investigation revealed that the ethanol extract derived from this herb showcased noteworthy anti-complement activity. Specifically, it targeted C1q, thereby impeding the classical pathway and the alternative pathway. This finding positions it as a promising contender for the role of potent anticomplement agents. Within China, *Taraxacum mongolicum Hand.-Mazz.*, a constituent of the Asteraceae family, holds eminence as a renowned medicinal plant [138]. It is often harnessed in addressing viral infections and inflammatory maladies. From this herb, a uniform water-soluble polysaccharide has been extracted. Mechanistic analyses have revealed that this compound curtails complement activation through the impediment of C1q. This trait renders it of significance in the context of managing conditions attributed to excessive activation of the complement system. Such actives currently express an inhibitory effect on C1q via the complement pathway but have not been specifically studied in the CNS and could be further investigated subsequently.

*Ganoderma lucidum*, a medicinal fungus, finds clinical utilization across numerous Asian countries as a means to bolster health and foster longevity [139]. Studies have shown that Ganoderma lucidum has neuroprotective effects, and aqueous extracts of Ganoderma lucidum significantly attenuate Aβ-induced synaptotoxicity by protecting synaptophysin. Likewise, the examination unveiled that *Ganoderma lucidum* polysaccharides (GLP) elicit a decrease in pro-inflammatory cytokines provoked by LPS or Aβ while concurrently fostering the expression of anti-inflammatory cytokines in BV-2 and primary microglial cells. Moreover, GLP mitigates the migratory propensity of microglia linked to inflammation, curtails morphological modifications, and diminishes the likelihood of phagocytosis. Remarkably, it also substantially reduces the expression of C1q [140]. Kaempferol, an innate antioxidant found in vegetables and fruits integral to human nutrition, displays a noteworthy capacity. Specifically, when administered, it hampers the proteolytic activation of complement C3 protein and the consequent emergence of reactive A1 astrocytes triggered by NPA in the striatum and hippocampus. Furthermore, it thwarts the augmentation of NF-κB expression and the heightened secretion of cytokines IL-1α, TNFα, and C1q, all of which are associated with the formation of reactive A1 astrocytes. Beyond this, kaempferol administration also averts the exacerbated production of amyloid β peptides within the striatum and hippocampus [127]. Cellular senescence, recognized as a pivotal hallmark of aging, entails an irreversible cessation of the cell cycle and becomes expedited during the aging trajectory. Intriguingly, black ginseng, a derivative of fresh ginseng achieved through a cyclical procedure of steaming and drying carried out nine times, has emerged under the spotlight owing to its physiological advantages in counteracting reactive oxygen species, inflammation, and oncogenic processes [141]. These mechanisms are frequently implicated in the onset of aging. Black ginseng attenuates cellular senescence by downregulating complement C1q and β-catenin signaling and its downstream activator in the senescence pathway, p53-p21/p16, to downregulated age-related inflammatory genes, especially in the complement system. 

*Astragalus* polysaccharides are one of the key active components of *Astragalus* membranaceus [87]. Pharmacological investigations have demonstrated that *Astragalus* polysaccharides exhibit a diverse range of pharmacological impacts, encompassing anti-inflammatory, antitumor, and immune regulatory properties. *Astragalus* polysaccharides regulate the polarization of microglia from M1 to M2 phenotype, reduce the secretion of inflammatory factors IL-1α, TNF-α, and C1q, and inhibit the activation of A1 neurotoxic astrocytes, thus effectively inhibiting neuroinflammation and demyelination in experimental autoimmune encephalomyelitis.

Tanshinone, a prominent lipid-soluble constituent of *Salvia miltiorrhiza*, takes shape as a substantial active ingredient, notably as TanIIA [142]. Extensive research has unveiled its pharmacological effects, particularly in the realm of neuroprotection. In the context of the rat brain, TanIIA emerges as a safeguard against Aβ-induced inflammation-induced neuronal impairment. A gamut of neuroprotective attributes can be ascribed to TanIIA, encompassing the attenuation of overactive glial cell response and the inhibition of inflammatory mediators such as IL-1β, IL-6, C1q, C3c, and C3d. Simultaneously, C1q surfaces as a countermeasure against the toxicity induced by oligomeric forms of Aβ. Its early upregulation in the aftermath of injury, distinct from the coordinated induction of other complement components, facilitates the orchestration of a gene expression program that fosters neuroprotection. This program, in turn, exhibits the potential to shield against Aβ-related pathologies during the preclinical phases of AD and other neurodegenerative processes.

Salidroside, a bioactive constituent sourced from *Rhodiola rosea*, is currently under scrutiny as a promising therapeutic avenue for addressing ischemic stroke. Particularly noteworthy is its potential effectiveness in curtailing the inflammatory response in the context of cerebral ischemia-reperfusion injury (IRI), as evidenced by studies conducted within the 24 h timeframe following the occurrence of ischemic brain events [143]. In the wake of cerebral IRI, there emerges a prompt escalation in the accumulation of immunoglobulin M, mannose-binding lectin 2, and annexin IV on cerebral endothelial cells. This is accompanied by the induction of complement components C3 and C3a within 24 h post-IRI. Subsequently, at the 48 h mark, a substantial surge is observed in the complement component C1q. Salicin affected these proteins and reversed their changes after 24 h of IRI. Salidroside operates as a neuroprotective agent by curtailing the premature activation of the lectin pathway on cerebral endothelial cells and impeding the gradual activation of the classical pathway following cerebral IRI. This protracted neuroprotection seems to be linked, at least in part, to the elevated expression of genes associated with neuroplasticity. This enhanced gene expression is facilitated by the mitigation of complement activation [144]. In a similar vein, Salidroside plays a role in diminishing inflammation and neuronal impairment after middle cerebral artery occlusion and reperfusion. This effect is, in part, attributed to the inhibition of cerebral complement C3 activation. Furthermore, Salidroside’s impact on astrocytes and microglial BV2 cells following oxygen–glucose deprivation and subsequent restoration did not extend to influencing C1q, C2, or C3 levels [145]. Within human umbilical vein endothelial cells (HUVEC), Salidroside served to safeguard against the decline of CD46 and CD59 while concurrently mitigating the elevation of VCAM-1, ICAM-1, P-selectin, and E-selectin. These effects correlated with reduced LDH release and an enhanced Bcl-2/Bax ratio. Crucially, these protective outcomes of Salidroside manifested only in the context of oxygen–glucose restoration. 

Significant progress in the relationship between C1q and CNS has been made in recent years, suggesting that C1q and complement pathways are regarded as potential therapeutic targets for CNS disorders. People tend to opt for dietary choices as intervention measures in addressing these ailments. Importantly, our limited grasp of the intricate mechanisms underlying C1q and complement-mediated neurodegenerative conditions has contributed to a considerable rate of unsuccessful endeavors in formulating dietary interventions for CNS disorders. The summary of current natural products or food components regulating C1q in CNS is shown in Table 3. There is a growing need for further research to explore food components as a specific and effective intervention targeting complement-related neurodegenerative diseases.

## 7. Discussion and Conclusions

With the increasing prevalence of neurodegenerative diseases and the frequent occurrence of their complications, people are becoming increasingly interested in obtaining safe, active ingredients from natural foods to replace the medications used for neurodegenerative disease treatment. 

C1q plays an important role in the early stages of the disease by labeling and eliminating cellular debris and microbes, orchestrating immune responses, signaling “danger”, and then activating the complement pathway that rapidly reacts to defend the body against external pathogens. In addition, C1q plays a key role in the mechanism by which glial cells regulate synaptic pruning by activating the complement pathway that contributes to CNS development. C1q is not only involved in synaptic growth and development but also plays distinct roles in the diverse stages of neurodegenerative diseases. Neuronal and glial cell death, as well as impaired cognitive function because of aging or genetic mutations manifested in neurodegenerative diseases, have all been shown to be inextricably linked to C1q. Thus, taking full advantage of C1q and complement-related mechanisms in physiological and pathophysiological conditions of CNS has become a new strategy for improving human health. Although significant progress in complement pathways in CNS has been made over the last decade, there are several issues that require further investigation. For example, how does C1q tagged and located on synapses and then activate microglia phagocytosis? There is limited knowledge of the ligands on synapse interact with C1q. How may we balance the crucial role of C1q in apoptotic cell debris clearance and synapse loss? Additionally, since complement inhibitors are ineffective against C1q, inhibitors and activators targeting C1q in vivo need to be discovered in further studies.

Currently, few studies have been conducted on C1q and brain health in the food industry. Studies that focus on daily food component-mediated C1q and complement-related diseases have a lot of untapped potential. Studies on the function of diverse dietary components related to C1q on CNS disorders are valuable for the clinical application prospects for disease intervention and control, especially in memory improvement and brain health in AD and others. Absolutely, various functional food factors could exert a neuroprotective role against brain injury by regulating C1q, like *Ganoderma lucidum* polysaccharides and Kaempferol. However, the specific underlying mechanisms have not been fully explained. Therefore, further studies are needed in the future to investigate the mechanism of targeting C1q to delay neurodegenerative diseases by key food functional components in the daily diet.

## Figures and Tables

**Figure 1 foods-12-03580-f001:**
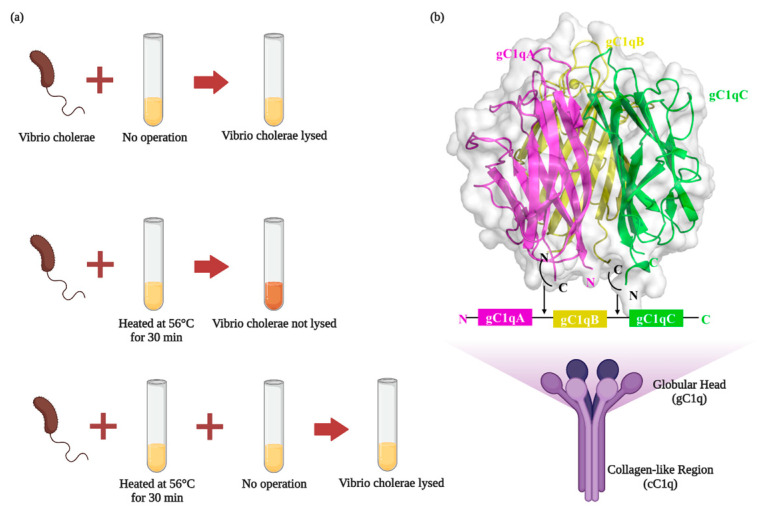
(**a**) Complementation findings. Sheep anti-cholera serum was able to lyse Vibrio cholera, while sheep anti-cholera serum lost the ability to lyse Vibrio cholera when heated at 56 °C for 30 min, and Vibrio cholera was again lyzed when fresh non-immune-serum was added to the heated serum again. (**b**) The structure of C1q (Created with BioRender.com accessed on 28 April 2023).

**Figure 2 foods-12-03580-f002:**
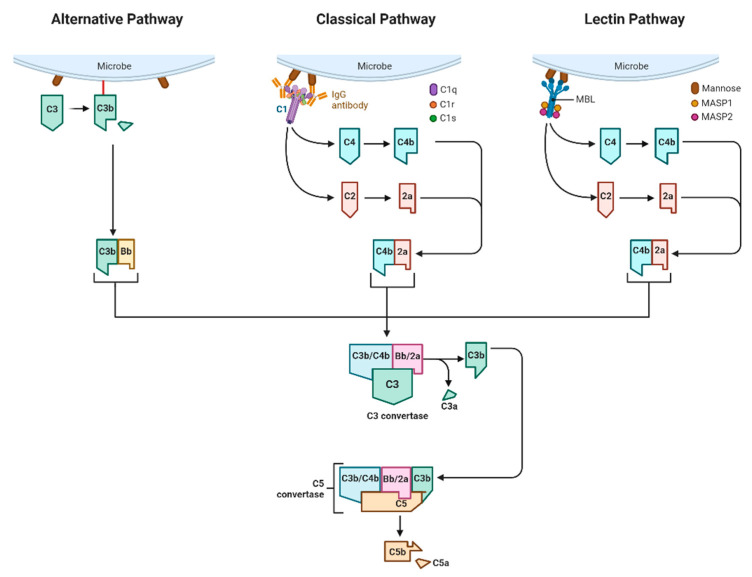
Complement pathways (created with BioRender.com accessed on 12 October 2022).

**Figure 3 foods-12-03580-f003:**
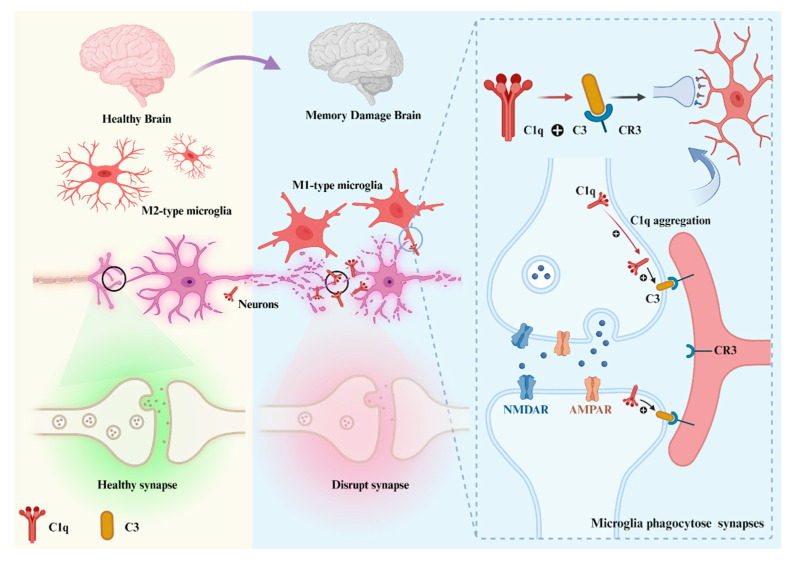
Microglia phagocytose synapses. C1q is expressed in microglia or neurons and localizes at synapses by recognizing ligands, leading to downstream deposition of complement protein C3, which binds to C3 receptors on microglia and activates phagocytosis of microglia to directly trigger synaptic loss (Created with BioRender.com accessed on 11 August 2023).

**Figure 4 foods-12-03580-f004:**
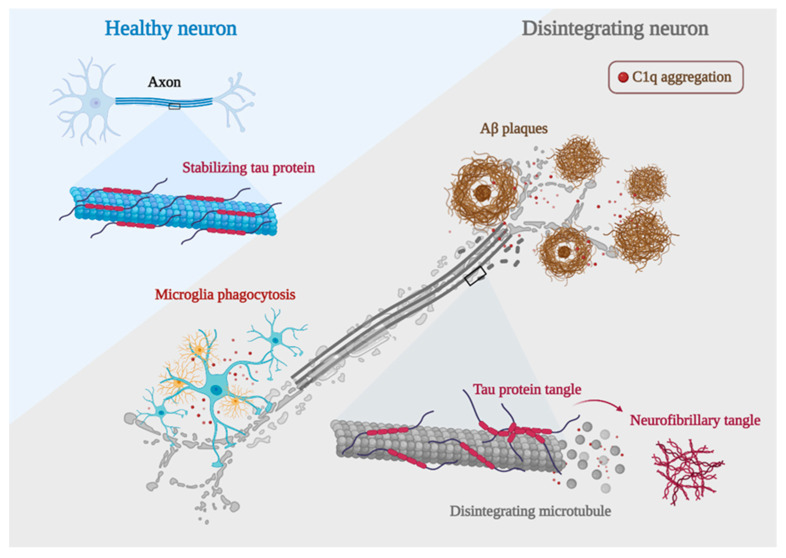
AD development mechanisms. Neurons with C1q aggregation show microglia engulfing healthy synapses, Aβ clumping forming plaques, hyperphosphorylated Tau proteins forming neurofibrillary tangles, and a significant decrease in their ability to bind microtubules due to Tau protein hyperphosphorylation, leading to microtubule disintegration and ultimately worsening of AD development (created with BioRender.com accessed on 12 October 2022).

**Figure 5 foods-12-03580-f005:**
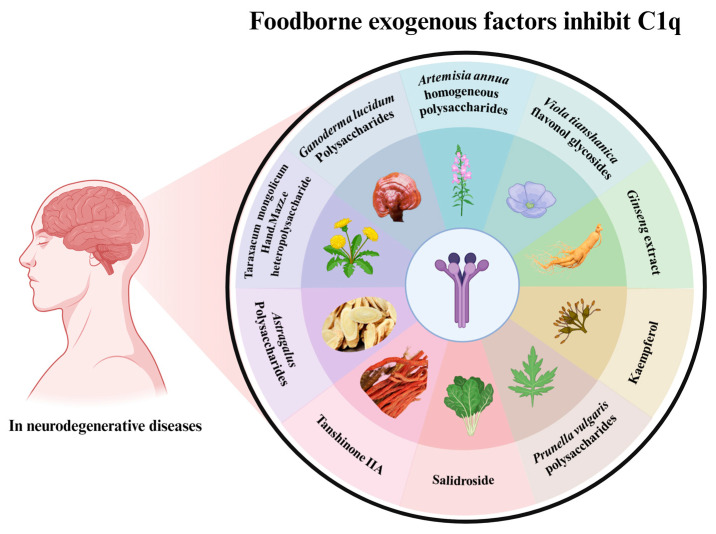
Inhibition of C1q aggregation by functional food factors (created with BioRender.com accessed on 28 May 2023).

**Table 2 foods-12-03580-t002:** Characteristics of neurodegenerative diseases.

Neurodegenerative Diseases	Animal/Cellular Model	Characteristics	C1q Effects	Ref.
AD	Tg2576 animals (APP) with C1q-deficient mice	Aβ plaques	Aβ co-localizes with C1q	[2,108,109]
	PS19 mice overexpressing the P301S mutant of human Tau/Hek cells	Tau protein misfolding and assembly	Tau protein co-localizes with C1q	[113,114,115]
	Tg2576 animals (APP) with C1q-deficient mice	Synapse elimination	Synapse co-localizes with C1q	[2,3,62]
PD	-	Depigmentation of the substantia nigra and locus coeruleus	C1q was restricted to microglia throughout the brain	[118]
	Autopsies from PD patients	Neuronal loss in the pars compacta of the substantia nigra		[8,120]
HD	Early HD patients	CAG trinucleotide repeat expansion in the huntingtin gene on chromosome 4	C1q produced locally by M1-type microglia is activated on the membranes of neurons	[124,134]
TBI	Sections of brains obtained at autopsy from 25 cases following closed TBI	Traumatic brain injury disrupts the BBB	C1q prompts the transformation of M2-type microglia into M1-type microglia and enhances complement system activation	[78,131]

**Table 3 foods-12-03580-t003:** Effect of functional food factors on C1q.

Sources	Main Active Ingredients	Mechanism of Action	Function	Ref.
*Artemisia annua* L.	Acidic homogeneous polysaccharides	Inhibited the classical pathway and the alternative pathway	Anti-complement activity	[135]
*Prunella vulgaris*	Homogeneous acidic polysaccharides	Reduced excessive activation of the complement system	Anti-complement activity	[136]
*Viola tianshanica flavonol glycosides*	Flavonol glycosides and other phenolic compounds	Inhibited the classical pathway and the alternative pathway	Anti-complement activity	[137]
*Taraxacum* *mongolicum Hand.-Mazz. heteropolysaccharide*	Heteropolysaccharide	Inhibited excessive activation of the complement system	Anti-complement activity	[138]
*Ganoderma lcidum*	Polysaccharides	Down-regulates LPS- or Aβ-induced pro-inflammatory cytokines, promotes anti-inflammatory cytokine expressions in BV-2 and primary microglia and reduces C1q expression	Neuroprotective	[140]
Black Ginseng	Panax ginseng	Downregulated age-related inflammatory genes, included in the complement system	Ameliorates cellular senescence	[141]
*Astragalus*	Polysaccharides	Regulates the polarization of microglia from M1 to M2 phenotype by inhibiting the miR-155, reduces the secretion of inflammatory factors, and inhibits the activation of neurotoxic astrocytes	Inhibit neuroinflammation and demyelination in experimental autoimmune encephalomyelitis	[87]
*Salvia miltiorrhiza*	Tanshinone IIA	Reduced the number of astrocytes and microglial cells and induced C1q decreased in the brain of Alzheimer’s disease model rats	Reduced inflammation levels of AD rats	[142]
*Rhodiola Rosea*	Salidroside	Reducing early activation of the lectin pathway on the cerebral endothelium and inhibiting the gradual activation of the classical pathway after cerebral IR	Neuroprotective	[144]
*Rhodiola Rosea*	Salidroside	Inhibited classical complement activation and increased CD46 and CD59	The protection afforded in cerebral ischemia-reperfusion injury	[145]
Vegetables and fruits	Kaempferol	Blocked the NPA-induced increase of NF-κB expression and enhanced secretion of cytokines IL-1α, TNFα, and C1q	Prevents the activation of complement C3 protein and the generation of reactive A1 astrocytes	[127]

## Data Availability

Data is contained within the article.

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
