# Peer review of "Complement in Human Brain Health: Potential of Dietary Food in Relation to Neurodegenerative Diseases"

_foods, 2023, doi:10.3390/foods12193580_

Round 1

Reviewer 1 Report

The review entitled "Complement in human brain health: potential of dietary food 2 in neurodegenerative diseases", conducted by Xing et., presents in a complex way the mechanisms of the C1q protein in the genesis of Alzheimer's disease. The manuscript is well described, featuring illustrations that facilitate reading.

The authors demonstrated a review that presents the importance of Complement in human brain health, associating this theme with a potential of dietary food in neurodegenerative diseases. It is an important topic for the field, which addresses a specific gap, which is the effect of the complement system on various neurodegenerative pathologies.  Most articles do not present in a generalized way the influence of the complement system on neurodegenerative diseases, as well as the mechanisms involved in this process. The manuscript addressed the suggested topic in depth and there are no recommendations for improvement.

Reviewer 2 Report

The authors, in this review, described and discussed literature conserning  the function and immunomodulatory mechanisms of C1q and its role on synaptic pruning in developing aging, or pathological conditions. They, then reported evidences on the relevance of C1q and the complement pathway in the atiopathogenetic mechanisms of neurodegenerative diseases, as well as on some foods with potential beneficial effects  via C1q and complement pathway. The results are interesting because permit to contribute to identify potential therapeutic targets for these pathologies, in alternative to pharmacological approaches. The review is well structured (event though sometimes redundant), clear and easy to read.

I have some issues to address: for instance, the acronyms: in the abstract section, insert at line 15, central nervous system (CNS).

I suggest to check for some minor Enghish errors: for instance, lines 468 and 762 (neurodegenerative diseases). 

Reviewer 3 Report

Manuscript #foods-2610318 is an interesting review on plants that contains anticomplement materials. In my humble opinion, the manuscript deserves consideration for publication but after revision to address some issues:

Errors in Figure 5 should be corrected. In some instances, the plant name is written (e.g. Viola tianshanica, Ginseng, …) while in others the active substance name is mentioned (Salidroside, kaempfero (maybe kaempferol), …). Also, revise Artemisia annuapoly and Artemisia annua polysaccharides! What is the basis to select and show these amongst those in Table 3. For example, Prunella vulgaris is not in the figure. All plants’ genus and species names should be italics. In addition, citations should come after the first mention. Please revise the mechanism of action for Artemisia annua in Table 3.

Additional issues:

The neurodegenerative Parkinson’s disease has in principle motor symptoms (Line 61).

What would this review benefit from listing amino acids residues (Lines 105-110).

The letter “B” should be capital in “b-chain” (Line 122).

Minor editing might enhance the manuscript language.

Round 2

Reviewer 3 Report

In my humble opinion, current revised version might be accepted for publication.